# Fundamental Limits of Budget-Fidelity Trade-off in Label Crowdsourcing

**Farshad Lahouti**
Electrical Engineering Department, California Institute of Technology
lahouti@caltech.edu

**Babak Hassibi**
Electrical Engineering Department, California Institute of Technology
hassibi@caltech.edu

## Abstract

Digital crowdsourcing (CS) is a modern approach to perform certain large projects using small contributions of a large crowd. In CS, a taskmaster typically breaks down the project into small batches of tasks and assigns them to so-called workers with imperfect skill levels. The crowdsourcer then collects and analyzes the results for inference and serving the purpose of the project. In this work, the CS problem, as a human-in-the-loop computation problem, is modeled and analyzed in an information theoretic rate-distortion framework. The purpose is to identify the ultimate fidelity that one can achieve by any form of query from the crowd and any decoding (inference) algorithm with a given budget. The results are established by a joint source channel (de)coding scheme, which represent the query scheme and inference, over parallel noisy channels, which model workers with imperfect skill levels. We also present and analyze a query scheme dubbed $k$-ary incidence coding and study optimized query pricing in this setting.

## 1 Introduction

Digital crowdsourcing (CS) is a modern approach to perform certain large projects using small contributions of a large crowd. Crowdsourcing is usually used when the tasks involved may better suite humans rather than machines or in situations where they require some form of human participation. As such crowdsourcing is categorized as a form of human-based computation or human-in-the-loop computation system. This article examines the fundamental performance limits of crowdsourcing and sheds light on the design of optimized crowdsourcing systems.

Crowdsourcing is used in many machine learning projects for labeling of large sets of unlabeled data and Amazon Mechanical Turk (AMT) serves as a popular platform to this end. Crowdsourcing is also useful in very subjective matters such as rating of different goods and services, as is now widely popular in different online rating platforms and applications such as Yelp. Another example is if we wish to classify a large number of images as suitable or unsuitable for children. In so-called citizen research projects, a large number of –often human deployed or operated– sensors contribute to accomplish a wide array of crowdsensing objectives, e.g., [2] and [3].

In crowdsourcing, a taskmaster typically breaks down the project into small batches of tasks, recruits so-called workers and assigns them the tasks accordingly. The crowdsourcer then collects and analyzes the results collectively to address the purpose of the project. The worker's pay is often low or non-existent. In cases such as labeling, the work is typically tedious and hence the workers usually handle only a small batch of work in a given project. The workers are often non-specialists

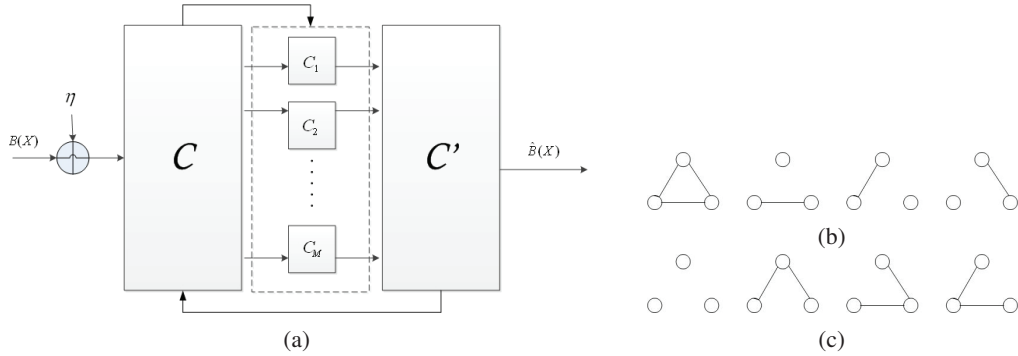

Figure 1: (a): An information theoretic crowdsourcing model; (b) 3IC for $N = 2$ valid responses, (c) invalid responses

and as such there may be errors in their completion of assigned tasks. Due to the nature of task assignment by the taskmaster, the workers and the skill level are typically unknown a priori. In case of rating systems, such as Yelp, there is no pay for regular reviewers but only non-monetary personal incentives; however, there are illegal reviewers who are paid to write fake reviews. In many cases of crowdsourcing, the ground truth is not known at all. The transitory or fleeting characteristic of workers, their unknown and imperfect skill levels and their possible motivations for spamming makes the design of crowdsourcing projects and the analysis of the obtained results particularly challenging.

Researchers have studied the optimized design of crowdsourcing systems within the setting described for enhanced reliability. Most research reported so far is devoted to optimized design and analysis of aggregation and inference schemes and possibly using redundant task assignment. In AMT-type crowdsourcing, two popular approaches for aggregation are namely majority voting and the Dawid and Skene (DS) algorithm [5]. The former sets the estimate based on what the majority of the crowd agrees on, and is provably suboptimal [8]. Majority voting is susceptible to error, when there are spammers in the crowd, as it weighs the opinion of everybody in the crowd the same. The DS algorithm, within a probabilistic framework, aims at joint estimation of the workers' skill levels and a reliable label based on the data collected from the crowd. The scheme runs as an expectation maximization (EM) formulation in an iterative manner. More recent research with similar EM formulation and a variety of probabilistic models are reported in [9, 12, 10]. In [8], a label inference algorithm for CS is presented that runs iteratively over a bipartite graph. In [1], the CS problem is posed as a so-called bandit survey problem for which the trade offs of cost and reliability in the context of worker selection is studied. Schemes for identifying workers with low skill levels is studied in, e.g., [6]. In [13], an analysis of the DS algorithm is presented and an improved inference algorithm is presented. Another class of works on crowdsourcing for clustering relies on convex optimization formulations of inferring clusters within probabilistic graphical models, e.g., [11] and the references therein.

In this work, a crowdsourcing problem is modeled and analyzed in an information theoretic setting. The purpose is to seek ultimate performance bounds, in terms of the CS budget (or equivalently the number of queries per item) and a CS fidelity, that one can achieve by any form of query from the workers and any inference algorithm. Two particular scenarios of interest include the case where the workers' skill levels are unknown both to the taskmaster and the crowdsourcer, or the case where the skill levels are perfectly estimated during inference by the crowdsourcer. Within the presented framework, we also investigate a class of query schemes dubbed $k$-ary incidence coding and analyze its performance. At the end, we comment on an associated query pricing strategy.

## 2    Modeling Crowdsourcing

In this Section, we present a communication system model for crowdsourcing. The model, as depicted in Figure 1a, then enables the analysis of the fundamental performance limits of crowdsourcing.

## 2.1 Data Set: Source

Consider a dataset $\mathcal{X} = \{X_1, \ldots, X_L\}$ composed of $L$ items, e.g., images. In practice, there is certain function $B(X) \in \mathcal{B}(\mathcal{X})$ of the items that is of interest in crowdsourcing and is here considered as the source. The value of this function is to be determined by the crowd for the given dataset. In the case of crowdsourced clustering, $B(X_i) = B_j \in \mathcal{B}(\mathcal{X}) = \{B_1, \ldots, B_N\}$ indicates the bin or cluster to which the item $X_i$ ideally belongs. We have $B(X_1, \ldots, X_n) = B(X^n) = (B(X_1), \ldots, B(X_n))$. The number of clusters, $|\mathcal{B}(\mathcal{X})| = N$, may or may not be known a priori.

## 2.2 Crowd Workers: Channels

The crowd is modeled by a set of parallel noisy channels in which each channel $C_i, i = 1, \ldots, W$, represents the $i$th worker. The channel input is a query that is designed based on the source. The channel output is what a user perceives or responds to a query. The output may or may not be the correct answer to the query depending on the skill level of the worker and hence the noisy channel is meant to model possible errors by the worker.

A suitable model for $C_i$ is a discrete channel model. The channels may be assumed independent, on the basis that different individuals have different knowledge sets. Related probabilistic models representing the possible error in completion of a task by a worker are reviewed in [8]. Formally, a channel (worker) is represented by a probability distribution $P(v|u), u \in \mathcal{U}, v \in \mathcal{V}$, where $\mathcal{U}$ is the set of possible responses to a query and $\mathcal{V}$ is the set of choices offered to the worker in responding to a query. For the example of images suitable for children, in general we may consider a shade of possible responses to the query, $\mathcal{U}$, including the extremes of totally suitable and totally unsuitable; As possible choices offered to the worker to answer the query, $\mathcal{V}$, we may consider two options of suitable and unsuitable. As described below, in this work we consider two channel models representing possibly erroneous responses of the workers: an $M$-ary symmetric channel model (MSC) and a spammer-hammer channel model (SHC).

An MSC model with parameter $\epsilon$, is a symmetric discrete memoryless channel without feedback [4] and with input $u \in \mathcal{U}$ and output $v \in \mathcal{V}$ ($|\mathcal{U}| = |\mathcal{V}| = M$), that is characterized by the following transition probability

$$P(v|u) = \begin{cases} \frac{\epsilon}{M-1} & v \neq u \\ 1 - \epsilon & v = u. \end{cases} \tag{1}$$

If we consider a sequence of channel inputs $u^n = (u_1, \ldots, u_n)$ and the corresponding output sequence $v^n$, we have $P(v^n|u^n) = \prod_{i=1}^n P(v_i|u_i)$, which holds because of the memoryless and no feedback assumptions. In case of clustering and MSC, the probability of misclassifying any input from a given cluster in another cluster only depends on the worker and not the corresponding clusters.

In the spammer-hammer channel model with the probability of being a hammer of $q$ (SHC($q$)), a spammer randomly and uniformly chooses a valid response to a query, and a hammer perfectly answers the query [8]. The corresponding discrete memoryless channel model without feedback, with input $u \in \mathcal{U}$ and output $v \in \mathcal{V}$ and state $C \in \{S, H\}$, $P(C = H) = q$ is described as follows

$$P(v|u, C) = \begin{cases} 0 & C = H \ and \ v \neq u \\ 1 & C = H \ and \ v = u \\ \frac{1}{|\mathcal{V}|} & C = S \end{cases} \tag{2}$$

where $C \in \{S, H\}$ indicates whether the worker (channel) is a spammer or a hammer. In the case of our current interest $|\mathcal{U}| = |\mathcal{V}| = M$, and we have $P(v^n|u^n, C^n) = \prod_{i=1}^n P(v_i|u_i, C^i)$.

In the sequel, we consider the following two scenarios: when the workers' skill levels are unknown (SL-UK) and when it is perfectly known by the crowdsourcer (SL-CS). In both cases, we assume that the skill levels are not known at the taskmaster (transmitter).

The presented framework can also accommodate other more general scenarios of interest. For example, the feedforward link in Figure 1a could be used to model a channel whose state is affected by the input, e.g., difficulty of questions. These extensions remain for future studies.

## 2.3 Query Scheme and Inference: Coding

In the system model presented in Figure 1a, encoding shows the way the queries are posed. A basic query is that the worker is asked of the value of $B(X)$. In the example of crowdsourcing for labeling images that are suitable for children, the query is "This image suits children; true or false?" The decoder or the crowdsourcer collects the responses of workers to the queries and attempts to infer the right label (cluster) for each of the images. This is while the collected responses could be in general incomplete or erroneous.

In the case of crowdsourcing for labeling a large set of dog images with their breeds, a query may be formed by showing two pictures at once and inquiring whether they are from the same breed [11]. The queries in fact are posed as showing the elements of a binary incidence matrix, $\mathbf{A}$, whose rows and columns correspond to $X$. In this case, $\mathbf{A}(X_1, X_2) = 1$ indicates that the two are members of the same cluster (breed) and $\mathbf{A}(X_1, X_2) = 0$ indicates otherwise. The matrix is symmetric and its diagonal is $1$. We refer to this query scheme as Binary Incidence Coding. If we show three pictures at once and ask the user to classify them (put the pictures in similar or distinct bins); it is as if we ask about three elements of the same matrix, i.e., $\mathbf{A}(X_1, X_2), \mathbf{A}(X_1, X_3)$ and $\mathbf{A}(X_2, X_3)$ (Ternary Incidence Coding). In general, if we show $k$ pictures as a single query, it is equivalent to inquiring about $C(k, 2)$ (choose 2 out of $k$ elements) entries of the matrix ($k$-ary Incidence Coding or $k$IC). As we elaborate below, out of the $2^{C(k,2)}$ possibilities, a number of the choices remain invalid and this provides an error correction capability for $k$IC.

Figures 1b and 1c show the graphical representation of 3IC, and the choices a worker would have in clustering with this code. The nodes denote the items and the edges indicate whether they are in the same cluster. In 3IC, if $X_1$ and $X_2$ are in the same cluster as $X_3$, then all three of them are in the same cluster. It is straightforward to see that in 3IC and for $N = 2$, we only have four valid responses (Figure 1b) to a query as opposed to $2^{C(3,2)} = 8$. The first item in Figure 1c is invalid because there are only two clusters ($N = 2$) (in case we do not know the number of clusters or $N \geq 3$, then this would remain a valid response). In this setting, the encoded signal $u$ can be one of the four valid symbols in the set $\mathcal{U}$; and similarly what the workers may select $v$ (decoded signal over the channel) is from the set $\mathcal{V}$, where $\mathcal{U} = \mathcal{V}$. As such, since in $k$IC the obviously erroneous answers are removed from the choices a worker can make in responding to a query, one expects an improved overall CS performance, i.e., an error correction capability for $k$IC. In Section 4, we study the performance of this code in greater details. Note that in clustering with $k$IC ($k \geq 2$) described above, the code would identify clusters up to their specific labellings.

While we presented $k$IC as a concrete example, there may be many other forms of query or coding schemes. Formally, the code is composed of encoder and decoder mappings:

$$\mathcal{C} : \mathcal{B}(\mathcal{X})^n \to \mathcal{U}^{\sum_{i=1}^{W} m_i}, \qquad \mathcal{C}' : \mathcal{V}^{\sum_{i=1}^{W} m_i} \to \hat{\mathcal{B}}(\mathcal{X})^n, \qquad (3)$$

where $n$ is the block size or number of items in each encoding (we assume $n | L$), and $m_i$ is the number of uses of channel $C_i$ or queries that worker $i, 1 \leq i \leq W$, handles. In many practical cases of interest, we have $\hat{\mathcal{B}}(\mathcal{X}) = \mathcal{B}(\mathcal{X})$ and we may have $n = L$. The rate of the code is $R = \sum_{i=1}^{W} m_i / n$ queries per item. In this setting, $\mathcal{C}'(\mathcal{C}(B(X^n))) = \hat{B}(X^n)$.

Depending on the availability of feedback from the decoder to the encoder, the code can adapt for optimized performance. The feedback could provide the encoder with the results of prior queries. We here focus on non-adaptive codes in (3) that are static and remain unchanged over the period of crowdsourcing project. We will elaborate on code design in Section 3.

Depending on the type of code in use and the objectives of crowdsourcing, one may design different decoding schemes. For instance, in the simple case of directly inquiring workers about the function $B(X_i)$, with multiple queries for each item, popular approaches are majority voting, and EM style decoding due to Dawid and Skene [5], where it attempts to jointly estimate the workers' skill levels and decode $B(X)$. In the case of clustering with 2IC, an inference scheme based on convex optimization is presented in [11].

The rate of the code is proportional to the CS budget and we use the rate as a proxy for budget throughout this analysis. However, since different types of query have different costs both financially (in crowdsourcing platforms) and from the perspective of time or effort it takes from the user to process it, one needs to be careful in comparing the results of different coding schemes. We shall elaborate on this issue for the case of $k$IC in Appendix E.

## 2.4 Distortion and the Design Problem

In the framework of Figure 1a, we are interested to design the CS code, i.e., the query and inference schemes, such that with a given budget a certain CS fidelity is optimized. We consider the fidelity as an average distortion with respect to the source (dataset). For a distance function $d(B(x), \hat{B}(x))$, for which $d(B(x^n), \hat{B}(x^n)) = \frac{1}{n} \sum_{i=1}^{n} d(B(x_i), \hat{B}(x_i))$, the average distortion is

$$
\begin{aligned}
D(B(X), \hat{B}(X)) &= Ed(B(X^n), \hat{B}(X^n)) \\
&= \sum_{X^n} P(B(X^n)) P(\hat{B}(X^n)|B(X^n)) d(B(X^n), \hat{B}(X^n)),
\end{aligned}
\tag{4}
$$

where $P(B(X^n)) = P(B(X))^n$ for iid $B(X)$. The design problem is therefore one of CS fidelity-query budget optimization (or distortion-rate, $D(R)$, optimization) and may be expressed as follows

$$
D^*(R^t) = \min_{\mathcal{C}, \mathcal{C}', R \leq R^t} D(B(X), \hat{B}(X))
\tag{5}
$$

where $R^t$ is a target rate or query budget. The optimization is with respect to the coding and decoding schemes, the type of feedback (if applicable), and query assignment and rate allocation. The optimum solution to the above problem is referred to as the distortion-rate function, $D^*(R^t)$ (or CS fidelity-query budget function). A basic distance function, for the case where $B(X)$ is discrete, is $l_0(B(X), \hat{B}(X))$, or the Hamming distance. In this case, the average distortion $D(B(X), \hat{B}(X))$ reflects the average probability of error. As such, the $D(R)$ optimization problem may be rewritten as follows

$$
D^*(R^t) = \min_{\mathcal{C}, \mathcal{C}', R \leq R^t} P(\mathsf{E} : \hat{B}(X) \neq B(X)).
\tag{6}
$$

In case of crowdsourcing for clustering, this quantifies the performance in terms of the overall probability of error in clustering. For other crowdsourcing problems, we may consider other distortion functions. Equivalently, we may consider minimizing the rate subject to a constrained distortion in crowdsourcing. The $R(D)$ problem is expressed as follows

$$
R^*(D^t) = \min_{\mathcal{C}, \mathcal{C}', D(B(X), \hat{B}(X)) \leq D^t} R = \min_{\mathcal{C}, \mathcal{C}', D(B(X), \hat{B}(X)) \leq D^t} \sum_{i=1}^{W} m_i / n
\tag{7}
$$

where $D^t$ is a target distortion or average probability of error. The optimum solution to the above problem is referred to as the rate-distortion function, $R^*(D^t)$ (CS query budget-fidelity function). In case, the taskmaster does not know the skill level of the workers, different users -disregarding their skill levels- would receive the same number of queries ($m_i = m', \forall i$); and the code design involves designing the query and inference schemes.

## 3 Information Theoretic CS Budget-Fidelity Limits

In the CS budget-fidelity optimization problem in (5), the code providing the optimized solution indeed needs to balance two opposing design criteria to meet the target CS fidelity: On one hand the design aims at efficiency of the query and making as small number of queries as possible; On the other hand, the code needs to take into account the imperfection of worker responses and incorporate sufficient redundancy. In information theory (coding theory) realm, the former corresponds to source coding (compression) and the latter corresponds to channel coding (error control coding) and coding to serve both purposes is a joint source channel code.

In this Section, we first present a brief overview on joint source channel coding and related results in information theory. Next, we present the CS budget-fidelity function in two cases of SL-UK and SL-CS described in Section 2.2.

### 3.1 Background

Consider the communication of a random source $Z$ from a finite alphabet $\mathcal{Z}$ over a discrete memoryless channel. The source is first processed by an encoder $\mathcal{C}$ and whose output is communicated over the channel. The channel output is processed by a decoder $\mathcal{C}'$, which reconstructs the source as $\hat{Z} \in \hat{\mathcal{Z}}$ and we often have $\mathcal{Z} = \hat{\mathcal{Z}}$.

From a rate-distortion theory perspective, we first consider the case where the channel is error free. The source is iid distributed with probability mass function $P(Z)$ and based on Shannon's source coding theorem is characterized by a rate-distortion function,

$$R^*(D^t) = \min_{\mathcal{C}, \mathcal{C}': D(Z, \hat{Z}) \leq D^t} I(Z, \hat{Z}), \tag{8}$$

where $I(.,.)$ indicates mutual information between two random variables. The source coding is defined by the following two mappings:

$$\mathcal{C}: \mathcal{Z}^n \to \{1, \ldots, 2^{nR}\}, \qquad \mathcal{C}': \{1, \ldots, 2^{nR}\} \to \hat{\mathcal{Z}}^n \tag{9}$$

The average distortion is defined in (4) and $D^t$ is the target performance. The optimization in source coding with distortion is with respect to the source coding or compression scheme, that is described probabilistically as $P(\hat{Z}|Z)$ in information theory. The proof of the source coding theorem follows in two steps: In the first step, we prove that any rate $R \geq R^*(D^t)$ is achievable in the sense that there exists a family (as a function of $n$) of codes $\{\mathcal{C}_n, \mathcal{C}'_n\}$ for which as $n$ grows to infinity the resulting average distortion satisfies the desired constraint. In the second step or the converse, we prove that any code with rate $R < R^*(D^t)$ results in an average distortion that violates the desired constraint. This establishes the described rate-distortion function as the fundamental limit for lossy compression of a source with a desired maximum average distortion.

From the perspective of Shannon's channel coding theorem, we consider the source as an iid uniform source and the channel as a discrete memoryless channel characterized by $P(V|U)$, where $U \in \mathcal{U}$ is the channel input and, $V \in \mathcal{V}$ is the channel output. The channel coding is defined by the following two mappings:

$$\mathcal{C}: \{1, \ldots, |\mathcal{Z}|\} \to \mathcal{U}^n \qquad \mathcal{C}': \mathcal{V}^n \to \{1, \ldots, |\mathcal{Z}|\} \tag{10}$$

The theorem establishes the capacity of the channel as $C = \max_{\mathcal{C}, \mathcal{C}'} I(Z, \hat{Z})$ and states that for a rate $R$, there exists a channel code that provides a reliable communication over the noisy channel if and only if $R \leq C$. Again the proof follows in two steps: First, we establish achievability, i.e., we show that for any rate $R \leq C$, there exists a family of codes (as a function of length $n$) for which the average probability of error $P(\hat{Z} \neq Z)$ goes to zero as $n$ grows to infinity. Next, we prove the converse, i.e., we show that for any rate $R > C$, the probability of error is always greater than zero and grows exponentially fast to $1/2$ as $R$ goes beyond $C$. This establishes the described capacity as the fundamental limit for transmission of an iid uniform source over a discrete memoryless channel.

For the problem of our interest, i.e., the transmission of an iid source (not necessarily uniform) over a discrete memoryless channel, the joint source channel coding theorem, aka source-channel separation theorem, is instrumental. The theorem states that in this setting a code exists that can facilitate reconstruction of the source with distortion $D(Z, \hat{Z}) \leq D^t$ if and only if $R^*(D^t) < C$. For completeness, we reproduce the theorem form [4] below.

**Theorem 1** *Let $Z$ be a finite alphabet iid source which is encoded as a sequence of $n$ input symbols $U^n$ of a discrete memoryless channel with capacity $C$. The output of the channel $V^n$ is mapped onto the reconstruction alphabet $\hat{Z}^n = \mathcal{C}'(V^n)$. Let $D(Z^n, \hat{Z}^n)$ be the average distortion achieved by this joint source and channel coding scheme. Then distortion $D$ is achievable if and only if $C > R^*(D^t)$.*

The proof follows a similar two step approach described above and assumes large block length ($n \to \infty$). The result is important from a communication theoretic perspective as a concatenation of a source code, which removes the redundancy and produces an iid uniform output at a rate $R > R^*(D^t)$, and a channel code, which communicates this reliably over the noisy channel at a rate $R < C$, can achieve the same fundamental limit.

## 3.2 Basic Information Theoretic Bounds

We here consider crowdsourcing within the presented framework, and derive basic information theoretic bounds. Following Section 2.1, we examine the case where a large dataset $\mathcal{X}$ ($L \to \infty$) and a function of interest $B(X)$ with associated probability mass function, $P(B(X))$, are available.

We consider the MSC worker pool model described in Section 2.2, where the skill set of workers are from a discrete set $\mathcal{E} = \{\epsilon_1, \epsilon_2, \ldots, \epsilon_{W'}\}$ with probability $P(\epsilon), \epsilon \in \mathcal{E}$. The number of workers in each skill level class is assumed large. We here study the two scenarios of SL-UK and SL-CS.

At any given instance, a query is posed to a random worker with a random skill level within the set, $\mathcal{E}$. We assume there is no feedback available from the decoder (non-adaptive coding) and the queries do not influence the channel probabilities (no feedforward). Extensions remain for future work.

The following theorem identifies the information theoretic minimum number of queries per item to perform at least as good as a target fidelity in case the skill levels are not known (SL-UK). The bound is oblivious to the type of code used and serves as an ultimate performance bound.

**Theorem 2** *In crowdsourcing for a large dataset of $N$-ary discrete source $B(X) \sim P(B(X))$ with Hamming distortion, when a large number of unknown workers with skill levels $\epsilon \in \mathcal{E}, \epsilon \sim P(\epsilon)$ from an MSC population participate (SL-UK), the minimum number of queries per item to obtain an overall error probability of at most $\hat{\epsilon}$, is given by*

$$R_{min} = \begin{cases} \frac{H(B(X)) - H_N(\hat{\epsilon})}{\log_2 M - H_M(E(\epsilon))} & \hat{\epsilon} \leq \min\{1 - p_{max}, 1 - \frac{1}{N}\} \\ 0 & otherwise, \end{cases} \tag{11}$$

*in which $H_N(\epsilon) \triangleq H(1 - \epsilon, \epsilon/(N-1), \ldots, \epsilon/(N-1))$, and $p_{max} = \max_{B(X) \in \mathcal{B}(\mathcal{X})} P(B(X))$.*

The proof is provided in Appendix A. Another interesting scenario is when the crowdsourcer attempts to estimate the worker skill levels from the data it has collected as part of the inference. In case this estimation is done perfectly, the next theorem identifies the corresponding fundamental limit on the crowdsourcing rate. The proof is provided in Appendix B.

**Theorem 3** *In crowdsourcing for a large dataset of $N$-ary discrete source $B(X) \sim P(B(X))$ with Hamming distortion, when a large number of workers with skill levels $\epsilon \in \mathcal{E}, \epsilon \sim P(\epsilon)$ -known to the crowdsourcer (SL-CS)- from an MSC population participate, the minimum number of queries per item to obtain an overall error probability of at most $\hat{\epsilon}$, is given by*

$$R_{min} = \begin{cases} \frac{H(B(X)) - H_N(\hat{\epsilon})}{\log_2 M - E(H_M(\epsilon))} & \hat{\epsilon} \leq \min\{1 - p_{max}, 1 - \frac{1}{N}\} \\ 0 & otherwise. \end{cases} \tag{12}$$

Comparing the results in Theorems 2 and 3 the following interesting observation can be made. In case the worker skill levels are unknown, the CS system provides an overall work quality (capacity) of an average worker; whereas when the skill levels are known at the crowdsourcer, the system provides an overall work quality that pertains to the average of the work quality of the workers.

## 4  $k$-ary Incidence Coding

In this Section, we examine the performance of the $k$-ary incidence coding introduced in Section 2.3. The $k$-ary incidence code poses a query as a set of $k \geq 2$ items and inquires the workers to identify those with the same label. In the sequel, we begin with deriving a lower-bound on the performance of $k$IC with a spammer-hammer worker pool. We then presents numerical results along with the information theoretic lower bounds presented in the previous Section.

### 4.1  Performance of $k$IC with SHC Worker Pool

We consider $k$IC for crowdsourcing in the following setting. The items $X$ in the dataset are iid with $N = 2$. There is no feedback from the decoder to the task manager (encoder), i.e., the code is non-adaptive. Since the task manager has no knowledge of the workers' skill levels, it queries the workers at the same fixed rate of $R$ queries per item. To compose a query, the items are drawn uniformly at random from the dataset. We assume that the workers are drawn from the SHC($q$) model elaborated in Section 2.2. The purpose is to obtain a lower-bound on the performance assuming an Oracle decoder that can perfectly identify the workers' skill levels (here a spammer or a hammer) and perform an optimal decoding. Specifically, we consider the following:

$$\min_{\mathcal{C}', \mathcal{C}:k\text{IC}} P(\mathsf{E} : \hat{B}(X) \neq B(X)) \tag{13}$$

where minimization is with respect to the choice of a decoder for a given $k$IC code. We note that the code length is governed by how the decoder operates, and often could be as long as the dataset. As

evident in (2), in the SHC model, the channel error rate (worker reliability) is explicitly influenced by the code and parameter, $k$. In the model of Figure 1a, this implies that a certain static feedforward exists in this setting. We first present a lemma, which is used later to establish a Theorem 4 on $k$IC performance. The proofs are respectively provided in Appendix C and Appendix D.

**Lemma 1** *In crowdsourcing for binary labeling ($N = 2$) of a uniformly distributed dataset, with $k$IC and a SHC worker pool, the probability of error in labeling of an item by a spammer ($C = S$), is given by*

$$\bar{\epsilon}_S = P(\mathsf{E} : \hat{B}(X) \neq B(X)|C = S) = \begin{cases} \frac{1}{k2^{k-1}} \sum_{i=0}^{\lfloor (k-1)/2 \rfloor} i \times \begin{pmatrix} k \\ i \end{pmatrix} & k \ \ odd \\ \frac{1}{k2^{k-1}} \left[ \sum_{i=0}^{\lfloor (k-1)/2 \rfloor} i \times \begin{pmatrix} k \\ i \end{pmatrix} + \frac{k}{4} \begin{pmatrix} k \\ k/2 \end{pmatrix} \right] & k \ \ even. \end{cases}$$

**Theorem 4** *Assuming crowdsourcing using a non-adaptive $k$IC over a uniformly distributed dataset ($k \geq 2$), if the number of queries per item, $R$, is less than $\frac{1}{k \ln(1-q)} \ln \frac{\hat{\epsilon}}{\bar{\epsilon}_S}$, then no decoder can achieve an average probability of labeling error less than $\hat{\epsilon}$ for any $L$ under the SHC($q$) worker model.*

To interpret and use the result in Theorem 4, we consider the following points: (i) The theorem presents a necessary condition, i.e., the minimum rate (budget) requirement identified here for $k$IC with a given fidelity is a lower bound. This is due to the fact that we are considering an oracle CS decoder that can perfectly identify the workers' skill levels and correctly label the item if the item is at least labeled by one hammer out of $R'$ times it is processed by the workers. (ii) In the current setting, where the taskmaster does not know the workers' skill levels, each item is included in exactly $R' \in \mathbb{Z}^+$ $k$-ary queries. That is due to the nature of the code $R'$. (iii) As discussed in Appendix E, Theorem 4 can also be used to establish an approximate rule of thumb for pricing. Specifically, considering two query schemes $k_1$IC and $k_2$IC, the query price $\pi$ is to be set as $\frac{\pi(k_1)}{\pi(k_2)} \approx \frac{k_1}{k_2}$.

## 4.2 Numerical Results

To obtain an information theoretic benchmark, the next corollary specializes Theorem 3 to the setting of interest in this Section.

**Corollary 1** *In crowdsourcing for binary labeling of a uniformly distributed dataset with a SHC($q$) worker pool -known to the crowdsourcer (SL-CS)- and number of choices in responding to a query of $M$, the minimum rate for any given coding scheme to obtain a probability of error of at most $\hat{\epsilon}$, is*

$$R_{min} = \begin{cases} \frac{1-H_b(\hat{\epsilon})}{q \log_2 M}, & 0 \leq \hat{\epsilon} \leq 0.5 \\ 0 & otherwise. \end{cases} \qquad queries \ per \ item \qquad (14)$$

Figure 2 shows the information theoretic limit of Corollary 1 and the bound obtained in Theorem 4. For rates (budgets) greater than the former bound, there exist a code which provides crowdsourcing with the desired fidelity; and for rates below this bound no such code exists. The coding theoretic lower bounds for $k$IC depend on $k$, $q$ and fidelity, and improve as $k$ and $q$ grow. The $k$IC bounds for $k = 1$ is equivalent to the analysis leading to Lemma 1 of [8].

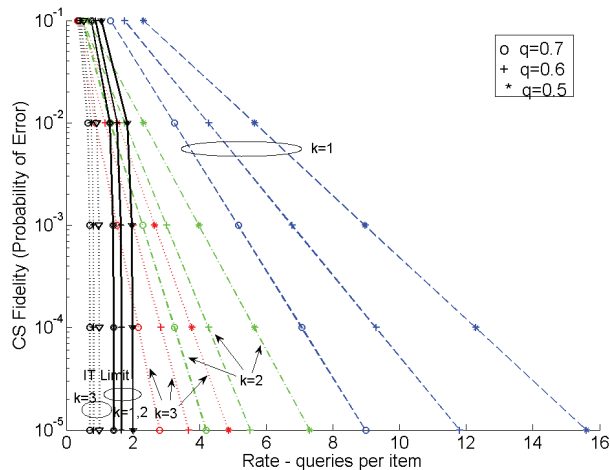

Figure 2: kIC performance bound and the information theoretic limit

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
