[Supplementary Material · CSIT-NIPS-Final-2.pdf]

**Appendixes**

## A Proof of Theorem 2

The parallel MSC channels representing workers with random skill levels may be viewed as a channel with random state [7]. The proof relies on the optimality of separate source and channel coding [7] (according to Shannon) in large dataset size and worker pool regime, which holds with a discrete memoryless source and discrete memoryless channel with random state. Here, we need a lossy source coder to bring the rate to $R(D(\hat{B}(X), B(X)) = \hat{\epsilon})$. If the source is memoryless, then we have

$$R(D(\hat{B}(X), B(X)) = \hat{\epsilon}) = \min I(B(X), \hat{B}(X)) = H(P(B(X))) - H(D = \hat{\epsilon}), \qquad (15)$$

where $H$ denotes the entropy and the minimization is with respect to the choice of a mapping (source coding scheme) $P(\hat{B}(X)|B(X))$ (disregarding the channel or worker possible errors). With Hamming distance, the second term amounts to $H_N(\hat{\epsilon})$, when $\hat{\epsilon} \leq \min\{1 - p_{max}, 1 - \frac{1}{N}\}$. If $\hat{\epsilon} \geq 1 - p_{max}$, then we can achieve the desired average distortion with rate zero and by a decoder that simply outputs argmax$P(B(X))$. Again, if $\hat{\epsilon} \geq 1 - \frac{1}{N}$ then we can achieve the desired average distortion with rate zero and by a decoder that uniformly outputs a $B(X)$ at random. We need the channel to be able to provide reliable communication for such a rate. In case the skill levels are unknown to both the taskmaster and the crowdsourcer, we have

$$nR(D = \hat{\epsilon}) \leq \sum_i^W m_i C_{SL-UK}. \qquad (16)$$

and

$$C_{SL-UK} = \max_{P(u)} I(U; V) = \log_2 M - H_M(E(\epsilon)) \qquad (17)$$

where the maximum occurs when $P(u) = 1/M$ and $E(.)$ denotes the expectation operator. Note that since the taskmaster does not know about the skill levels, it can only communicate the queries to workers with an identical rate. Combining (15), (16) and (17), the proof is complete. Note that the optimality of the separate source and channel coding scheme invoked here relies on proving an achievable scheme and a converse, which is the basis for the latter part of the Theorem statement. This can be done in line with what is presented in sections 3.9 and 7.4 of [7] and is omitted for brevity.

## B Proof of Theorem 3

The proof follows the same steps as in that of Theorem 2. However, the capacity when the skill levels are known at the crowdsourcer is given by

$$C_{SL-CS} = \max_{P(u)} I(U; V|\epsilon) = \log_2 M - E(H_M(\epsilon)) \qquad (18)$$

where the maximum occurs when $P(u) = 1/M$. Note again that since the taskmaster does not know about the skill levels, it can only communicate the queries to workers with an identical rate.

## C Proof of Lemma 1

The probability of error, when the query is posed to a spammer in a SHC, can be quantified as follows,

$$
\begin{aligned}
P(\mathsf{E}|C = \mathsf{S}) \quad &= P(\mathsf{E} : \hat{B}(X) \neq B(X)|C = \mathsf{S}) \\
&= \sum_{u \in \mathcal{U}} \sum_{v \in \mathcal{V}} P(u, v|C = \mathsf{S}) P(\mathsf{E}|U = u, V = v, C = \mathsf{S}) \\
&= \sum_{u \in \mathcal{U}} \sum_{v \in \mathcal{V}} P(u) P(v|u, C = \mathsf{S}) P(\mathsf{E}|U = u, V = v, C = \mathsf{S}) \\
&= \sum_{u \in \mathcal{U}} \sum_{v \in \mathcal{V}} \frac{1}{M^2} P(\mathsf{E}|U = u, V = v)
\end{aligned}
\qquad (19)
$$

The last equality follows in part because of the uniform distribution of the dataset ($P(B(X)) \sim$ uniform) and that $|\mathcal{U}| = |\mathcal{V}| = M$.

Each $k$IC symbol can be represented by a vector of length $k$, whose elements are in the set $\{0, 1, \ldots, N-1\}$ (here $N = 2$). As such, for $k$IC with arbitrary $k \geq 2$ and $N = 2$, the number of valid responses to a query can be counted as

$$k' = \begin{cases} \sum_{i=0}^{\lfloor (k-1)/2 \rfloor} \binom{k}{i} & k \text{ odd} \\ \sum_{i=0}^{\lfloor (k-1)/2 \rfloor} \binom{k}{i} + 0.5 \binom{k}{k/2} & k \text{ even} \end{cases} \tag{20}$$

or alternatively $k' = 2^{k-1}$. This is obtained noting that for $k \geq 2$, $k$IC does not recover specific labellings (e.g., in clustering it does put items in their associated bins but does not label the bins). In the same direction, each query of $k$IC that is posed to a worker, is equivalent to transmission of a symbol over a $k'$-ary discrete channel ($M = k'$), or alternatively a codeword of $k$ bits over an equivalent binary discrete channel. As such, any linear combination of two codewords over the corresponding field would create another codeword (up to labeling). As a result,

$$\sum_{u \in \mathcal{U}} \sum_{v \in \mathcal{V}} P(\mathsf{E}|U = u, V = v) = \frac{k'}{k} \sum_{u' \in \mathcal{U}} w_H(u')$$

where $w_H$ is the Hamming weight. Noting (20) and setting $M = k'$ in (19) the proof is complete.

## D   Proof of Theorem 4

Under the SHC model, we consider an oracle decoder who only makes a mistake on task $i$ if it is only assigned to spammers. Formally, the average error probability at the decoder, if in all $R'$ queries per task (codeword) only spammers are selected, is given by

$$P(\mathsf{E} : \hat{B}(X) \neq B(X)) = \sum_{\mathbf{C}} P(\mathsf{E} : \hat{B}(X) \neq B(X)|C_1, \ldots, C_{R'})P(C_1, \ldots, C_{R'}) =$$

$$P(\mathsf{E}|\mathbf{C} = \mathbf{S}) \times (1-q)^{R'} = P(\mathsf{E}|C = \mathsf{S}) \times (1-q)^{R'}$$

The last equality follows, since when the decoder observes $R'$ instances of spammers, it does not have any more information as observing a single spammer. In $k$IC, the number of queries per item $X$, $R$, is given by $R = \frac{1}{k}R'$. As such, we have the following result for an arbitrary decoder

$$\hat{\epsilon} = P(\mathsf{E} : \hat{B}(X) \neq B(X)|C = \mathsf{S}) \times (1-q)^{kR} \tag{21}$$

Using Lemma 1, the desired result is obtained. Note that as evident in the lemma $\bar{\epsilon}_S = P(\mathsf{E}|C = \mathsf{S})$ is also a function of $k$.

## E   Pricing Strategy

Using Equation (21) (in proof of) Theorem KICBound, one can draw some insights into the design of a CS system based on $k$IC. Below we consider this in the context of pricing queries. Specifically, in crowdsourcing with $k$IC query scheme with a spammer-hammer worker distribution of hammer probability $q$, we consider two scenarios: (i) the case where the price of the query per item is fixed at $\pi$ and does not change with $k$ and (ii) the case where the query price is a function of $k$, $\pi(k)$. In the former case, one can readily examine that since $P(\mathsf{E}|C = \mathsf{S})$ in Lemma 1 grows very slowly with $k$, any increase in $k$ directly translates to a smaller rate $R$ and hence crowdsourcing cost $\pi nR$ for a given crowdsourcing fidelity $\hat{\epsilon}$. In the latter case, however, the analysis sheds light on the appropriate pricing range. Specifically, for a given crowdsourcing budget and two query schemes $k_1$IC and $k_2$IC with rates $R_1$ and $R_2$, we have $\pi(k_1)R_1 = \pi(k_2)R_2$; and for a given $\hat{\epsilon}$ from (21), we have $k_1 R_1 \approx k_2 R_2$. This indicates that the pricing in the two scenarios compare as $\frac{\pi(k_1)}{\pi(k_2)} \approx \frac{k_1}{k_2}$. This sets a threshold (range) for crowdsourcing pricing strategy: If we are to use a $k_2$IC as opposed to $k_1$IC ($k_2 > k_1$), then we would have to pay at most $\pi(k_2) \lesssim \frac{\pi(k_1)k_2}{k_1}$. Note that due to said nature of $P(\mathsf{E}|C = \mathsf{S})$, this approximation is more accurate for larger values of $k_1$ and $k_2$ and when their difference is small.