[Reviews · NeurIPS 2016]

Reviewer 1

Summary

The paper models the problem of labeling data from crowdsourcing. Each worker is modeled as a noisy channel (two types of channels are considered: symmetric memoryless channel without feedback and spammer/hammer channel model). The problem then becomes a coding problem for the channel considered and the authors study. The budget used is then mapped on the rate of the code for the analysis. The paper considers distortion vs. budget tradeoffs and also studies a basic coding scheme (termed k-incidence coding) which corresponds to requiring the crowdsourced tasks to be that of labeling a subset of k items at a time.

Qualitative Assessment

I found the paper fairly interesting, but as it falls outside of my area of expertise don't have any detailed comments on it.

Confidence in this Review

1-Less confident (might not have understood significant parts)


Reviewer 2

Summary

The authors provide an information-theoretic treatment of the crowdsourcing problem, and obtain fundamental limits on the number of queries per item (in an asymptotic sense) under such a model. More specifically, the crowdsourcing problem is modeled as an information-theoretic problem of joint source channel coding (JSCC) with random channel state. The channel state is assumed to be unknown at the encoder, but may or may not be known at the decoder. The labels of the items are modeled as an i.i.d. source; the taskmaster is modeled as an encoder; the crowd workers are modeled as multiple uses of a channel; and the workers’ skill levels are modeled as random states of the channel; and the crowdsourcer is modeled as the decoder. With this model, the authors use existing results on the coding rate in the JSCC problem to establish the fundamental limits of the asymptotic number of queries per item in crowdsourcing. The authors also provide a lower bound on the error probability of a coding scheme called “k-ary incidence coding” under a specific crowd-sourcing model (or channel under their information theoretic model).

Qualitative Assessment

In summary, this is a well-written paper, and may be a first step toward the JSCC modeling of the crowdsourcing problem. Some technical comments: 1. The crowd-sourcing problem is modeled as JSCC with random channel state. The authors simply use the standard source-channel separation theorem in information theory to obtain their results (Theorem 2 and 3). However, the authors should first argue that the source-channel separation theorem still holds when the channel has a random state, where the state is unknown at the encoder, but may or may not be known at the receiver. I believe the separation theorem still holds in these situations, but the authors should prove it or find an existing proof. 2. The taskmaster is modeled as an encoder who only sees the labels of the items, and the crowdsourcer is modeled as a decoder who only sees the labels produced by the workers. However, this model may be too restrictive, because it ignores the fact that both the taskmaster and the crowdsourcer may have access to the original data (the items). This knowledge of the original data can provide side information for the encoder and decoder in the JSCC problem. The side information may not be helpful if it is only present at the encoder, but it can be very helpful if it is present at both the encoder and decoder or even at the decoder alone. I would have felt the model to be more convincing and realistic if the authors had considered side information in the model. 3. In Sec 4.2, when commenting on the curves in Fig 2, the authors say that the lower bounds for kIC improve as k and q grow. I guess the authors mean that the lower bound becomes smaller as k and q grow, which means that the performance of the code may get better. This however does not mean that the lower bound itself improves, because there is no upper bound to compare with. Edited following authors' response: The use of joint source-channel coding to model crowdsourcing is interesting, but (as another reviewer pointed out) a nonasymptotic characterization of the fundamental limits of CS would be more useful. The authors should take a look at recent results of Kostina and Verdu, pertaining to nonasymptotic converses for JSCC: V. Kostina and S. Verdu, "Lossy Joint Source-Channel Coding in the Finite Blocklength Regime", IEEE Trans. on Information Theory, vol. 59, no. 5, pp. 2545-2575, May 2013

Confidence in this Review

2-Confident (read it all; understood it all reasonably well)


Reviewer 3

Summary

The problem of aggregating worker output in crowdsourcing is addressed using the language and tools of information theory, and the fundamental limits of the budget-fidelity tradeoff are given.

Qualitative Assessment

I did not fully understand the paper (see below) but thought that the insights and model itself were a very interesting, and potentially useful, take on the design of crowdsourcing aggregation techniques. I would have liked to see the high-level intuition/insight stemming from these results (e.g. end of section 3) highlighted more prominently, as well as a conclusion for what one should take away from this (I know space is tight...) While I certainly appreciated the efforts of the authors to make the paper accessible to those not very familiar with information theory, like myself, in the end it was still quite hard to follow. I would recommend restating most/many of the results in terms of the crowdsourcing application itself, rather than retaining the language of channels and codes and distortion. Please mention in the related work the alternate (and slowly merging) line of research on robust economic incentives for workers (usually by the terms "crowdsourcing mechanism" or "mechanism design for crowdsourcing"). Some of this work also discusses varying worker abilities, and recently, how to combine these incentives with more sophisticated aggregation methods. Little things: 84: "is reviewed in" should be "are reviewed in"? 106: I was quite confused here (and earlier, in the initial explanation) about the distinction between the crowdsourcer and taskmaster, perhaps because "crowdsourcing" is a portmanteau of "crowd" and "outsourcing", which one would think would happen before workers got involved 122: type in "A A(X1,X2)" 275-276: "In case..." this was very hard to parse. figure 2: I could not glean much from this figure. Perhaps a more detailed caption would help.

Confidence in this Review

1-Less confident (might not have understood significant parts)


Reviewer 4

Summary

The paper considers a crowdsourcing set up where a taskmaster divides a big task (say an inference problem) into small tasks. These small tasks are then presented to crowd or workers. A crowdsourcer then collects the information provided by these workers (after completion of their assigned tasks) and process this information to conclude the inference task. This setup allows for the worker to have varying skill levels as well as to be unreliable in reporting their information to the crowdsourcer. The objective of the whole system is to ensure certain level of fidelity or 'goodness' of the final outcome at the crowdsourcer, even in the presence of some wrong information presented by the participating workers. This requires the taskmaster to carefully design small tasks (or queries) to be allocated to the workers. The paper defines the total number of queries presented to the workers as the budget of the crowdsourcing system. The paper studies a fundamental trade-off between the budget (number of queries presented to the workers by the taskmaster) and fidelity (quality of the inference task performed at the crowdsourcer). This trade-off is obtained by mapping the crowdsourcing system described above to the problem of joint source channel coding in information theory literature. The taskmaster is treated as an transmitter and the crowdsourcer is mapped to a receiver/decoder. The unreliable links from the workers to the crowdsourcer constitute a communication channel. Given this mapping a lower bound on the budget (no. of queries) for a desirable fidelity level follows from the standard results in information theory literature. This bound is independent of the schemes used by the taskmaster to design the queries and the algorithm used at the crowdsourcer to process the information collected from the workers. The authors then consider a specific scheme at the taskmaster, called k-ary incidence coding and analyze its performance. The authors then compare the performance of this encoding scheme with the obtained information theoretic lower bound for a specific channel model.

Qualitative Assessment

The reviewer finds the parallel drawn between the rate-distortion framework and the crowdsourcing system with unreliable workers quite interesting. This work may invite other researcher to study the crowdsourcing framework using information theoretic tools. However, the reviewer is not sure if the results presented in this paper model the real crowdsourcing systems in a very realistic way. One reason behind this could be that the paper wants the crowdsourcing problem to exactly map to the point-to-point communication problem so that the existing results from the information theory literature can be directly applied with the minimal efforts. It would be great if the authors can comment on the following issues. 1) It seems that the reliability and skill level of the workers is merged together and modeled as a single channel. The paper assumes that the transmitter (taskmaster) allocates same no. of queries to all the workers irrespective of their skill level. This may not be true in the real systems where the workers may carry some rating with them which represents their skill level. 2) How realistic are the channel models to take the unreliability of the workers into account. What about the workers which can have some knowledge of the task at hand and try to degrade the fidelity in an adversarial manner. The explanation of Numerical results is not very clear. The labeling of various curves in Fig.~2 can be improved. What does $k=1$ case correspond to. The worker directly gives you the correct label of the presented query as you can not present a pair and ask if they share the same label in this case? In page 6, line 246 the authors mention that they assume the knowledge of $P(B(X))$. The reviewer believe that this should have been stated much earlier in the paper. In page 4, line 135: Figure 1.a --> Figure 1.b In page 4, line 135: Figure 1.b --> Figure 1.c The authors repeatedly use 'Theorem', 'Lemma' and 'Corollary' even when these terms are not accompanied by a specific theorem or lemma number. The reviewer believes that 'theorem', 'lemma' and 'corollary' would be more appropriate in such cases.

Confidence in this Review

2-Confident (read it all; understood it all reasonably well)


Reviewer 5

Summary

The paper models digital crowdsourcing in an information theoretic framework to provide lower bounds on the needed number of queries per task to achieve a given probability of error. The workers are modeled as parallel independent channels. The type of channels considered are symmetric discrete memoryless channels (MSC) to model errors done by workers and spammer-hammer channel model SHC(q) to model the case when a worker is a spammer with probability 1-q. The code is assumed to be nonadaptive. Two cases are considered for analysis: the case when the crowdsourcer, who decides the value of each task after collecting the results from the workers, knows the skill level of all of the workers and when it do not know anything about them. In both cases the taskmaster, who distributes the tasks to the workers, is assumed to not know anything about the workers skills, so every worker is assigned the same number of tasks. Finally, a necessary condition on the minimum needed rate- number of queries per task- k-ary incidence coding, where the worker is shown k items and asked if they belong to similar or different breeds, with SHC(q) worker pool was provided.

Qualitative Assessment

Well organized and well written paper. There was no experiments or simulations to show how far current used schemes are from the limits, for example in AMT. I think section 2.4 can be shortened and 3.1 is not necessary or can be shortened. Instead, proof sketches to the theorems are more important. It seems novel, there is no analysis of the crowdsourcing problem from an information theoretic point of view before as far as I know and claimed by the paper. It seems very useful where designers of such crowdsourcing projects can know the minimum rate needed to accomplish a task with a given probability of error and it can help in pricing such projects. I did not read the appendix so I cannot claim anything about the proof.

Confidence in this Review

2-Confident (read it all; understood it all reasonably well)


Reviewer 6

Summary

This work provides an information-theoretic bound on budge-fidelity tradeoff for label crowdsourcing using rate-distortion theory and channel coding theorem.

Qualitative Assessment

The information-theoretic bound is well derived by this paper. If this is the first attempt that applied rate-distortion theory and channel coding theorem to the crowdsourcing problem, it can be a reasonably important contribution to the field. However, the novelty and practical values of this paper are still questionable, as the derivation of the information-theoretic bound is relatively straightforward and the bound has a risk of being very loose. The main problem is that unlike the channel coding problem, where we have a large amount of freedom in designing a codebook or codewords (we can use repetition code, Hamming code, convolutional code, RS code, low density parity check (LDPC) code), it is not clear whether we have such freedom in designing the sequence of channel input u^n, In equation (3), almost all mappings from B(X)^n to U^(\sum m_i) (except degenerate ones) should be valid or implementable codebooks, otherwise the channel capacity may be a loose bound. Obviously for crowdsourcing, something like repetition code is possible where the natural decoding method is majority voting. However, beyond this, it is unclear how we can design u^n to improve the error correcting capability. In particular, there is striking difference between channel coding problem and the crowdsourcing problem in that we cannot assign the same query to the same worker multiple times as this will ruin the iid assumption in equation (1) or (2), as the same worker will probably generate the same answer to the same query, whereas in communication channel we can assume the channel is iid and same query can be used multiple times for the same channel. This is just one example, and in general, it is very likely that the crowdsourcing problem has lots of fundamental constraints in equation (3) unlike channel coding, where an encoding mapping {0,1}^{nR} -> {0,1}^n has little constraint. Therefore, there should be important constraints on the set of possible encoding schemes in equation (3), and due to such constraints the proposed information-theoretic bound has a risk of being very loose. Finally, it would be better to compare the performances of state-of-the-art methods with the derived information-theoretic bound. Minor comments: - In (8) min is over C, C' with E[d(Z,\hat{Z}) ] \leq D^t - Proof for theorem 4 can be more elaborated. - In theorem 4, ln(1-q) < 0. Is \epsilon_S larger than \hat{\epsilon} ? - In (17), the reason why we have H_M(E(\epsilon)) should be explained or it needs some citation of the previous work.

Confidence in this Review

2-Confident (read it all; understood it all reasonably well)